# Charge decay in the spatial afterglow of plasmas and its impact on diffusion regimes

Nabiel H. Abuyazid [1] ✉, Necip B. Üner[2,3], Sean M. Peyres[2] & R. Mohan Sankaran [2] ✉

The spatial afterglow is a region at the boundary of a non-equilibrium plasma where charged species relax into ambient equilibrium. In many applications, the spatial afterglow is the part of the plasma that interacts with surfaces, such as suspended particles or a material substrate. However, compared to the bulk plasma, there has been little effort devoted to studying the properties of the spatial afterglow, and a fundamental analysis has not yet been developed. Here, we apply double Langmuir probe measurements and develop an advection-diffusion-recombination model to provide a detailed description of charged species in the spatial afterglow over a wide range of pressures, temperatures, plasma dimensions, and flow rates. We find that the density of charged species in the spatial afterglow decays by orders of magnitude, which leads to a transition from ambipolar to free diffusion. These insights can be used to explain or predict experimental observations of phenomena, such as the charging of dust grains and the dose of charged species to a biomaterial.

Plasmas are unique sources of charged species for technological applications such as ion-beam modification of materials[1–4], surface neutralization[5,6], thermal cooling via ionic wind[4,7], dust removal (electrostatic precipitation)[8], and plasma electrochemistry[9–14]. In general, the density and dynamics of charged species in the bulk volume of plasmas are relatively well-described by quasi-neutrality and ambipolar diffusion. In many applications, particularly those where the plasma is configured as a flow reactor or a jet, the properties of charged species at the boundaries of the plasma, such as the region that bridges the bulk plasma with the equilibrium background gas, are becoming increasingly important. For example, aerosol particles nucleated from vapor precursors[15,16], or introduced from the ambient or other sources[17], should be predominantly charged negatively inside a plasma because of the higher mobility of electrons. However, recent experimental[18–21] and computational[22,23] studies have revealed that aerosol particles leave a plasma with a bipolar charge distribution. To explain the shift in the charge distribution, a separate region outside the bulk plasma, termed the spatial afterglow, has been proposed, where particles could become neutralized or positively charged during transit[18,22,23]. There is now growing experimental evidence supporting the existence of a spatial afterglow that is distinct from the bulk plasma both in terms of a physical regime and properties such as the charging environment[17,18,24]. The spatial afterglow is also relevant to other applications such as those where a plasma jet interacts with a surface. In plasma medicine, therapeutic effects appear to depend on the dose of charged species delivered to a surface[25,26]. Similarly, in charge neutralization applications used in industrial settings, electrons and ions are used to eliminate electrostatic charge on surfaces[27–29]. Fundamental investigations have observed that charged species in a plasma jet help overcome Rayleigh and Kelvin-Helmholtz instabilities on liquid surfaces through electrohydrodynamic effects[30]. Underlying all of these examples is the density or flux of charged species at the surface, highlighting the need for a mathematical description to estimate the concentration profile of charged species in the spatial afterglow. However, an in-depth understanding of how the plasma decays in the spatial afterglow has not yet been established, and the potential change of fundamental properties, such as quasi-neutrality and ambipolar diffusion, between the bulk plasma and spatial afterglow remain unclear.

[1]Department of Chemical and Biomolecular Engineering, University of Illinois at Urbana-Champaign, Champaign, IL, USA. [2]Department of Nuclear, Plasma, and Radiological Engineering, University of Illinois at Urbana-Champaign, Champaign, IL, USA. [3]Chemical Engineering Department, Middle East Technical University, Ankara, Turkey. ✉e-mail: abuyazi2@illinois.edu; rmohan@illinois.edu

In general, the spatial afterglow is the spatial analog of the more well-known temporal afterglow[31], which describes the period when the plasma extinguishes after the power coupling is turned off, and there is a subsequent, rapid decay in the density of charged species. This decay leads to several interesting effects, first shown by Allis & Rose[32] and later by Phelps[33]. At the beginning of the temporal afterglow, the ambipolar field keeps ions and electrons in proximity of each other on the order of the Debye length. As the charged species density decreases due to diffusive loss and charge recombination, there is concurrent growth of the Debye length and weakening of the ambipolar field. Finally, the diffusive behavior of charged species transitions to free diffusion, i.e., thermal motion. Phelps calculated the apparent diffusion coefficient for electrons and positive ions as a function of a normalized length, defined as the ratio of the characteristic diffusion length and the Debye length, and showed three regimes: (1) an ambipolar diffusion regime at large values (> 1000 for typical plasmas), (2) a free diffusion regime at small values (< 1), and (3) a transition between these regimes at a critical value of ~30[33].

The spatial afterglow is distinct from the temporal afterglow because the power source and the bulk plasma are never turned off; the decay occurs in space (outside of the bulk plasma volume) rather than in time, and advection can play a key role (when there is a gas flow, such as in a jet). In applications, the spatial afterglow is experienced continuously; for example, in plasma jets, contact with a surface determines the dose of ions and electrons, which, in turn, increases with the duration of exposure. In plasma-aerosol systems, a steady-state flow carries particles into and out of the spatial afterglow, which determines their final charge and the extent of agglomeration[24,34,35]. To date, most studies of spatial afterglows have been motivated by their application as a source of low-energy ions for mass spectrometry, known as a selected ion flow tube (SIFT)[36], and have focused on understanding the formation, diffusion, and advection of large molecular ions[37]. These studies are not directly related to the spatial afterglow of plasmas operated in simple gases and at higher pressures. A very recent work provided a description of charged species densities in the spatial afterglow by adapting the framework for temporal afterglows; however, crucial charged species loss mechanisms, including

advective transport and three-body recombination, were not included[19]. Overall, despite their importance, there are far fewer fundamental studies of spatial afterglows than their temporal analogs.

Here, we present a combined experimental and modeling study of the spatial afterglow of a continuous-flow plasma operated over a range of pressures to elucidate the structure of the afterglow and provide a comprehensive analysis of the decay of charged species. Plasma parameters, including the electron temperature and plasma densities, were spatially measured by a double Langmuir probe (DLP). We then developed an advection-diffusion-recombination model to describe the charge decay in the spatial afterglow and found excellent quantitative agreement with experiments. Because DLP diagnostics in our experiments were limited to pressures below 300 Torr, the experimentally validated model allowed us to obtain insights all the way up to atmospheric pressure. The model reveals that, similar to a temporal afterglow, there is a transition from ambipolar to free diffusion, but in space rather than in time, corresponding with the spatial, as opposed to temporal, charge decay. Moreover, the transition in diffusive regimes occurs at different distances within the spatial afterglow depending on key variables including pressure, gas temperature, plasma dimension, and gas flow rate. These results provide key insights into applications of similar plasmas, such as the charging of aerosol particles and the dose of charged species to a surface.

## Results
### Characterization of the bulk plasma
An experimental setup was specially designed to characterize the spatial afterglow of a continuous-flow argon plasma (Supplementary Fig. 1). The setup consisted of the primary bulk plasma, which was formed inside a quartz tube, and the secondary spatial afterglow, which emanated into a grounded stainless-steel chamber. The plasma was operated over a wide range of pressures (10–300 Torr) radio frequency (RF) at a constant forward power of 20 W. Figure 1 shows photos of the bulk plasma between the powered (suspended ring) and ground (Ultra-Torr fitting) electrodes and the spatial afterglow (hereafter referred to as afterglow) at a series of operating pressures. As the pressure increases from 10 to 75 Torr, the bulk plasma looks visibly the

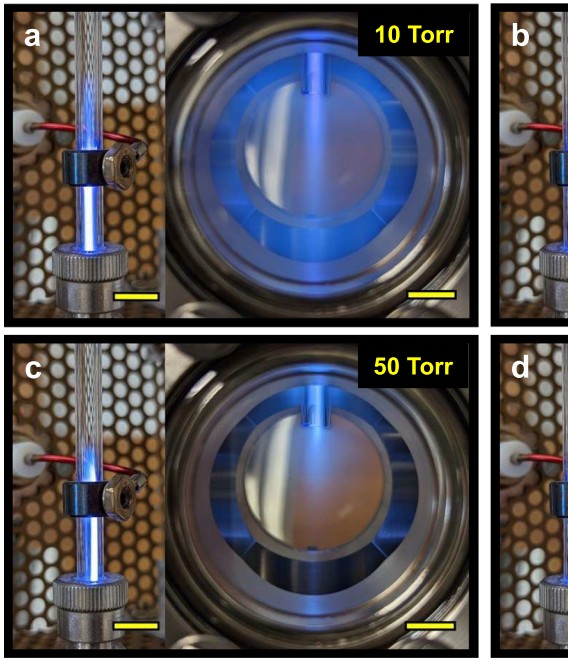
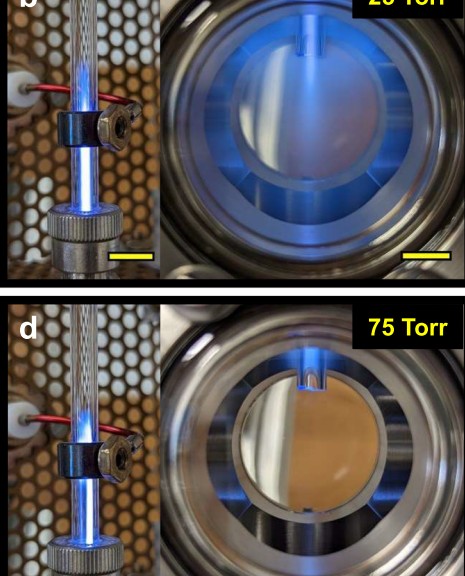

**Fig. 1 | Experimental setup for generation and characterization of spatial afterglow.** Photographs of a primary argon plasma and its spatial afterglow operating at (**a**) 10, (**b**) 25, (**c**) 50, and (**d**) 75 Torr. The forward RF power measured at the power supply was 20 W, and the argon flowrate was 1000 sccm in all cases. The scale bars correspond to 1 cm.

same, but the afterglow dramatically contracts with increasing pressure to virtually no visible afterglow at 75 Torr and above. As we will discuss below, while the visual change in the afterglow is related to the plasma density, the decrease in the visible intensity of the afterglow does not necessarily mean that there are no charged species present, only that the plasma density, and in particular, the density of radiating species, has decreased.

Because the properties of the afterglow depend on the properties of the bulk plasma, we initially characterized the bulk plasma. The plasma density in the bulk plasma, $n_p$, was obtained by measuring the current and voltage using an RF power probe and applying a fluid plasma model in combination with an equivalent circuit analysis (Supplementary Note 1 and Supplementary Fig. 2)[18,38,39]. Despite minor visual changes (Fig. 1), the plasma density in the bulk was found to increase linearly with pressure with a slope, $\beta = 5.41 \times 10^{17}\,\mathrm{m}^{-3}\,\mathrm{Torr}^{-1}$ (Fig. 2a).

## Characterization of spatial afterglow

We next characterized the afterglow, located downstream of the bulk plasma, by performing spatial measurements in the stainless steel chamber using a DLP (Supplementary Note 2 and Supplementary Figs. 3–5). Compared to single Langmuir probe (SLP), DLP has a floating circuit and hence, withdraws no net current, which is critical to minimizing perturbation of the plasma and in particular, the afterglow, where the density of charged species is low, and there is no electrical ground nearby. Moreover, at higher pressures, SLP can couple to the plasma and draw large currents, which leads to a filamentary discharge and prevents the formation of a afterglow (Supplementary Fig. 6). The DLP was translated axially along the radial centerline of the afterglow, and the afterglow density was measured as a function of distance from the bulk plasma at different pressures. During these measurements, the DLP caused no visible change to the afterglow. As shown in Fig. 2b, the afterglow density decreases somewhat monotonically with increasing distance at all pressures studied. We note that the zero point represents the physical position where the DLP was nearest to the ground electrode of the bulk plasma and increasing distance corresponds to the probe being moved further away from the bulk plasma. With increasing pressure, in stark contrast to the bulk plasma, the afterglow density exhibits a large drop between 75 and 150 Torr, with smaller changes between 10 and 75 Torr and between 150 and 300 Torr. Qualitatively, these results are explained by a net loss of charged species in the afterglow with increasing pressure and distance due to gas-phase collisions and diffusion, and unlike in the bulk plasma, continuous power coupling is not provided to maintain some degree of ionization.

## Modeling of the spatial afterglow

In support of experimental measurements, and to provide additional insight, a one-dimensional (1D) advection-diffusion-recombination model was developed and applied to the decay of charged species in the afterglow. The first step was to determine the bulk plasma–afterglow boundary with respect to the physical DLP positions in the experiments. We invoked a power dissipation model to define the bulk plasma volume, which could be related to the boundary. Briefly, the power coupled to the bulk plasma is dissipated volumetrically by elastic collisions, three-body recombination, and surface recombination on the walls. In our analysis, we found that the contribution of three-body recombination was negligible compared to elastic collisions in the volume, so it was ignored. Solving the model yielded the volume and therefore, the length of the bulk plasma. The bulk plasma length could then be defined as the sum of length of the gap between the powered and ground electrode and the length of the plasma beyond the tip of the ground electrode, i.e., plasma-afterglow boundary, $\gamma_0$. Qualitatively, with increasing pressure, $\gamma_0$ was found to decrease, indicating that the plasma becomes shorter; in other words,

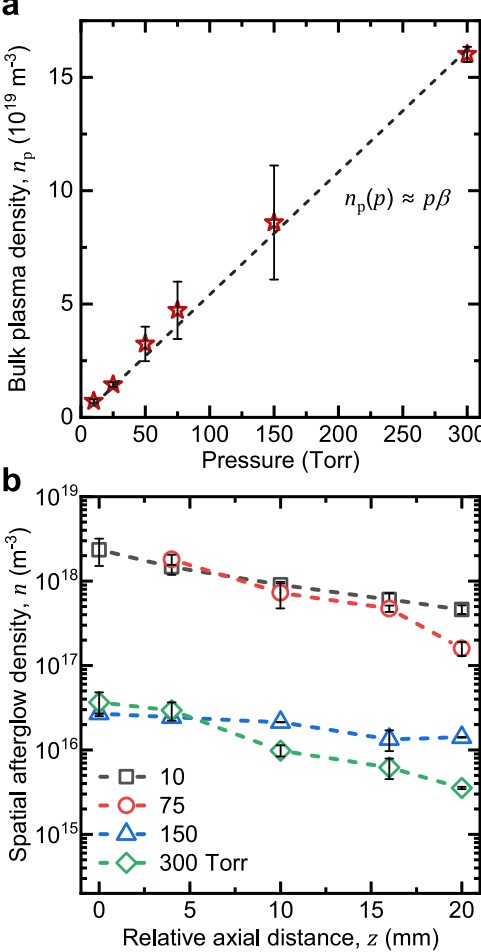

**Fig. 2 | Experimentally measured plasma densities in the bulk plasma and the spatial afterglow. a** Bulk plasma density, $n_p$, as a function of pressure obtained from equivalent circuit analysis and a one-dimensional plasma-fluid model, showing that $n_p$ increases linearly with pressure with a slope, $\beta = 5.41 \times 10^{17}\,\mathrm{m}^{-3}\,\mathrm{Torr}^{-1}$. **b** Spatial afterglow density as a function of axial distance, $n(z)$, obtained from DLP measurements. The zero point corresponds to the spatial position where the DLP was close to the ground electrode of the bulk plasma. At least four measurements were made at each of the five spatial positions, and error bars correspond to one standard deviation.

the plasma–afterglow boundary is located further upstream with pressure, as visually observed in Fig. 1. Defining the plasma–afterglow boundary, $\gamma_0$, achieves the following: (1) it delineates the point from where ionization is considered negligible, and (2) it defines the probe position in the diagnostic chamber with reference to the beginning of the afterglow as a function of pressure. A complete description of the power dissipation model is provided in the Supplementary Note 3.

Beyond the plasma-afterglow boundary, $\gamma_0$, charged species in the afterglow engage in advection (mediated by the neutral gas), radial diffusion, and volumetric three-body recombination. For an axisymmetric geometry and assuming quasi-neutrality and ambipolar diffusion, a radially-averaged advection-diffusion-reaction model expresses the one-dimensional charged species density, $n(z)$, in the spatial afterglow as:

$$n(z) = \frac{n_p}{\left(1 + \frac{k_r n_g n_p \Lambda^2}{2D_a}\right)\exp\left[\frac{2D_a}{v_f \Lambda^2}(z - \gamma_0)\right] - \frac{k_r n_g n_p \Lambda^2}{2D_a}} \quad (1)$$

where $k_r$ is the three-body recombination rate constant, $n_g$ is the neutral gas density, $\Lambda$ is a characteristic diffusion length in the radial

direction, and $v_f$ is the fluid velocity. Computational fluid dynamics suggested that $\Lambda$ is approximately equal to the tube radius (i.e., negligible jet expansion in the chamber, see Supplementary Note 4 and Supplementary Fig. 7), and, therefore, $v_f$ is approximately equal to the average fluid velocity in the tube. In this formulation, the origin of the $z$-axis is the ground electrode, and $\gamma_0$ is the distance to the bulk plasma-afterglow boundary. We note that all experiments were performed in Ar and to be consistent, the model also assumes only Ar as the background gas. Thus, ionization by mechanisms other than electron-impact such as Penning ionization were not considered. Additional details behind the derivation of Eq. 1 are presented in the Supplementary Note 5.

In addition, we assumed that electrons exhibit a Maxwellian energy distribution in the afterglow. Specifically, the Einstein relation was used to obtain the ambipolar diffusion coefficient, $D_a$, which is derived based on Maxwell-Boltzmann statistics. This assumption is self-consistent with our DLP analysis, where the orbital-limited electron flux expression was used. In support, we measured the electric fields in the bulk plasma and afterglow (Supplementary Note 6 and Supplementary Tables 1 and 2) and calculated EEDFs at various conditions using LoKI-B[40,41] (Supplementary Note 7). These calculations showed that the EEDF tended toward a Maxwellian at higher pressures and smaller values of the reduced electric field, $E/N$ (Supplementary Fig. 8). Previous experiments have shown that deviations from Maxwellian equilibrium are inversely proportional to the degree of ionization in the plasma[42], and calculations have shown that the solution to the Boltzmann equation for cooled electrons under low field conditions is a Maxwellian[43]. Thus, the assumption of a Maxwellian is especially reasonable in the afterglow, where electrons are lost and relax, and the electric field is low.

Qualitatively, Eq. 1 predicts a decreasing trend for the density of charged species in the afterglow with increasing distance from the bulk plasma. To quantitatively determine the charge density profiles, all parameters involved in Eq. 1 can be readily calculated or estimated using literature data; here, we relied on experimental measurements. The electron temperature, $T_e$, was measured as a function of axial position by DLP. Because $T_e$ did not vary significantly along the axial direction in the afterglow (Supplementary Fig. 4), it was assumed to be spatially uniform, and an average of the measurements was used (1.5 eV). The gas and ion temperatures were assumed to be room temperature (300 K). $\gamma_0$ was kept as a fitting parameter and then compared with our power model analysis (Supplementary Eqn. 18).

With all parameters of the model defined, we nondimensionalized Eq. 1 to obtain a more generalized description of the afterglow:

$$\eta(\xi) \equiv \frac{n}{n_p} = \frac{1}{\exp(\xi) + \phi^2[\exp(\xi) - 1]} \qquad (2)$$

where $\xi$ is the axial distance in the afterglow, $z - \gamma_0$, normalized to the characteristic decay length, $\lambda_c \equiv (v_f \Lambda^2/2 D_a)$, and $\phi \equiv (k_r n_g n_p \Lambda^2/2 D_a)^{1/2}$. From Eq. 2, we can define $\phi$ as a Thiele modulus, a well-known dimensionless parameter found in chemical engineering that relates the ratio of the rates of diffusion to reaction. Here, $\phi$ compares the relative rates of three-body recombination to diffusional loss of charge. We note that for plasmas formed in gas mixtures, the afterglow would have additional Thiele moduli for each reaction that leads to charged species annihilation or generation, such as ion attachment and Penning ionization.

We validated the spatial afterglow model by comparing with experimental measurements. Figure 3a shows that $\gamma_0$ obtained from DLP measurements agrees very well with calculations based on the power model over all the pressures studied. For the power model analysis, $\beta$ was determined from the charged species density in the bulk plasma (Fig. 2a). The forward power was directly measured, but for RF plasmas, it is known to differ, sometimes substantially, from the

power absorbed, $P_w$, which is difficult to assess[41]. For this reason, we assumed $P_w$ to be independent of pressure and was approximated to be the forward power (20 W).

We next calculated $\phi$ as a function of pressure, comparing calculations assuming a linear approximation for bulk plasma density, and calculations based on exact experimental measurements. The Thiele modulus is quite sensitive to changes in pressure, with a $p^{3/2}$ dependence, and the model and experiment show excellent agreement, as shown in Fig. 3b. When $\phi < 1$, diffusive losses dominate the charged species' decay in the afterglow, and when $\phi \geq 1$, three-body recombination becomes prominent. Physically, when the pressure is low, three-body recombination in the volume is limited, and diffusion to the walls is the primary loss mechanism for charged species, and when the pressure is high, three-body recombination is enhanced. These loss mechanisms balance each other when $\phi \approx 1$ at ~150 Torr.

Finally, the dimensionless charge density, $\eta$, was calculated as a function of the dimensionless axial distance in the afterglow, $\xi$, for different values of $\phi$ and was compared to DLP measurements (Fig. 2b), where corresponding spatial coordinates for DLP measurements are adjusted by $\gamma_0$ (Fig. 3a). As shown in Fig. 3c, at relatively low pressures (<75 Torr), the decay profile is found to follow an exponential trend and exhibited excellent agreement with experimental measurements. As the pressure rises, so does the density of neutral gas atoms, thus increasing the volumetric recombination rate, resulting in greater values of $\phi$. As shown in Fig. 3c, when $\phi \approx 1$ at 150 Torr, the decay profile shifts away from an exponential and develops curvature at small distances. At higher pressures (300 Torr), where $\phi \approx 2$, the shift is even more apparent. Model calculations and experimental measurements show good agreement for 300 Torr and moderately so for 150 Torr. Overall, charged species density measurements via DLP provide evidence for the validity of the proposed spatial afterglow model.

## Extending model calculations to other process conditions

The benefit of a working model is that the calculations can be extended to other process conditions, including those challenging for experiments. For example, atmospheric pressure is an important process condition for applications of continuous-flow plasmas. However, we were unable to perform conventional DLP analysis of the afterglow at pressures higher than 300 Torr. Specifically, at 400 Torr, the DLP traces did not exhibit the expected sigmoidal shape corresponding to the electron retardation region, and were instead linear, suggesting a low density of charged species and, in particular, the absence of hot electrons. Here, we define hot electrons as those with energies substantially higher than that of the background gas. Thus, the shift to a linear DLP trace could be interpreted as a transition to a free diffusion regime in which the hot electrons are rapidly lost from the afterglow. We expect that at higher pressures, the charge decays very rapidly with distance from the bulk plasma, and DLP measurements need to be made very close to the ground electrode, which is challenging with a physical probe. Furthermore, we were not able to create a stable discharge above 400 Torr, since a discharge at those pressures required much higher power density, which was not attainable when the forward power was fixed at 20 W. Interestingly, this was captured by our power model as $\gamma_0$ becomes negative for ~400 Torr, meaning that the plasma will not span between the powered and ground electrodes (Fig. 3a).

With our validation of the model at pressures where experimental measurements were possible, we are able to circumvent the experimental challenges faced at higher pressures by extending our model calculations to higher pressures up to atmospheric (~760 Torr). Input plasma parameters for the model were obtained by extrapolating Fig. 2a to 760 Torr for $n_p$ ($4.1 \times 10^{20}$ m$^{-3}$), and assuming the same $T_e$ (1.5 eV) measured by DLP at lower pressures, which was found to be consistent with other reports of continuous-flow, atmospheric-pressure plasmas with similar geometries to that studied here[15,18,38,44–47].

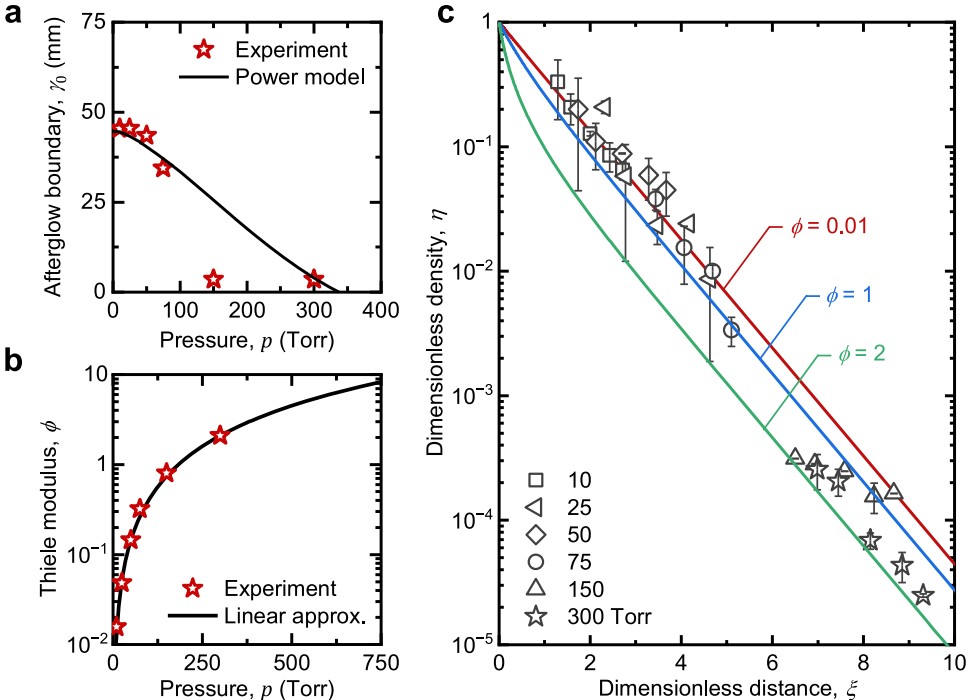

**Fig. 3 | Model calculations of spatial afterglow properties. a** Location of bulk plasma–afterglow boundary, $\gamma_0$, as a function of pressure, calculated using a power dissipation model (solid black line). $\gamma_0$ values obtained by fitting the model to DLP measurements are also shown for comparison (red star data points). **b** Thiele modulus, $\phi$, as a function of pressure, calculated using the approximate linear relationship for the bulk plasma density (solid black line). Values of $\phi$ reflecting experimental conditions are also shown for comparison (red star data points).

**c** Dimensionless charge density, $\eta$, as a function of dimensionless axial distance, $\xi$, in the spatial afterglow for different values of $\phi$ calculated using an advection-diffusion-recombination model (solid lines). Experimental values obtained by DLP measurements are also shown for comparison (data points). At least four measurements were made at each of the five spatial positions, and error bars correspond to one standard deviation.

An important issue that must be considered as the pressure increases is that the gas temperature in the plasma will increase as a result of the increasing electron–neutral collision frequency. We note that because the charge density decreases in the afterglow, the temperature in the plasma will not be the same as in the afterglow, and it may decrease substantially depending on various conditions already discussed, such as pressure and distance, as well as other parameters, including gas flow rate and geometry. To address this issue, we developed a 1D heat transfer model for the bulk plasma to estimate the gas temperature at the end of the bulk plasma, i.e., at the plasma–afterglow boundary (Supplementary Note 8). Briefly, the model includes mechanisms for heat generation, such as electron–neutral elastic collisions and recombination, in the volume as well as at the walls, including heat loss by radiation and natural convection to the walls. We then assume two limiting conditions for the temperature in the afterglow. The first is isothermal, where the temperature in the afterglow is the same as the plasma–afterglow boundary and represents an upper bound. The second is room temperature (300 K), where the temperature in the spatial afterglow is assumed to rapidly decrease to the ambient and represents a lower bound. The actual temperature in the afterglow could be between these upper and lower bounds and will depend on the specific system; for this reason, we have purposefully left our analysis general.

Figure 4a shows advection-diffusion-recombination model calculations for the decay of charged species at 760 Torr at an upper bound and lower bound temperature, where the former was calculated by the heat transfer model to be 1470 K. Results for 300 Torr are also shown for comparison where the upper bound temperature was calculated to be 680 K. We note that below 300 Torr, the upper bound temperature was found to be close to room temperature, consistent with results shown in Fig. 3c. Focusing on the room temperature profiles, the Thiele modulus, $\phi$, increases from $\phi \approx 0.1$ at 75 Torr to

$\phi \approx 20$ at 760 Torr, confirming that, as expected from our previous analysis (Fig. 3b), three-body recombination becomes predominant at higher pressures. This change in the dominant loss mechanisms leads to a very steep decay in density, a nearly thousand-fold decrease, within distances on the order of millimeters near the bulk plasma. This rapid decay also highlights the decreasing $\gamma_0$ at higher pressures (Supplementary Eq. 18). We also observe that because the upper bound temperature increases with pressure, there is a larger possible span of charge density profiles (Supplementary Table 3 and Supplementary Fig. 9), but we note that gas temperature has the effect of reducing the decay of charge, serving to counter the effect of pressure.

The model developed also facilitates the study of the effect of other process conditions on the decay of charged species in the afterglow, including gas flow rate and bulk plasma dimension. Much like pressure, these conditions will also affect the gas temperature in the bulk plasma. We can once again apply the heat transfer model and perform our analysis at an upper and lower bound temperature. Figure 4b, c show model calculations for a selection of gas flow rates and inner tube diameters, respectively, while keeping other parameters fixed such as the pressure (atmospheric). Increasing the gas flow rate from 500 to 2500 sccm appears to generally extend the afterglow because of the enhanced advective transport which overcomes diffusional losses (decreasing $\lambda_c$) (Fig. 4b). Concomitantly, a higher gas flow rate leads to a lower bulk plasma temperature as a result of forced convection (Supplementary Fig. 10), which decreases the span of charge density profiles as compared to lower gas flow rates. This model result agrees with previous observations that high gas flow rates are needed to form an afterglow jet in an atmospheric-pressure plasma setup[48,49]. The tube diameter affects the relative rate of three-body recombination to diffusional losses (increasing $\phi$), without affecting the relative rate of diffusional losses to advective transport (constant $\lambda_c$) (Supplementary Fig. 11). Decreasing the tube diameter enhances heat transfer and

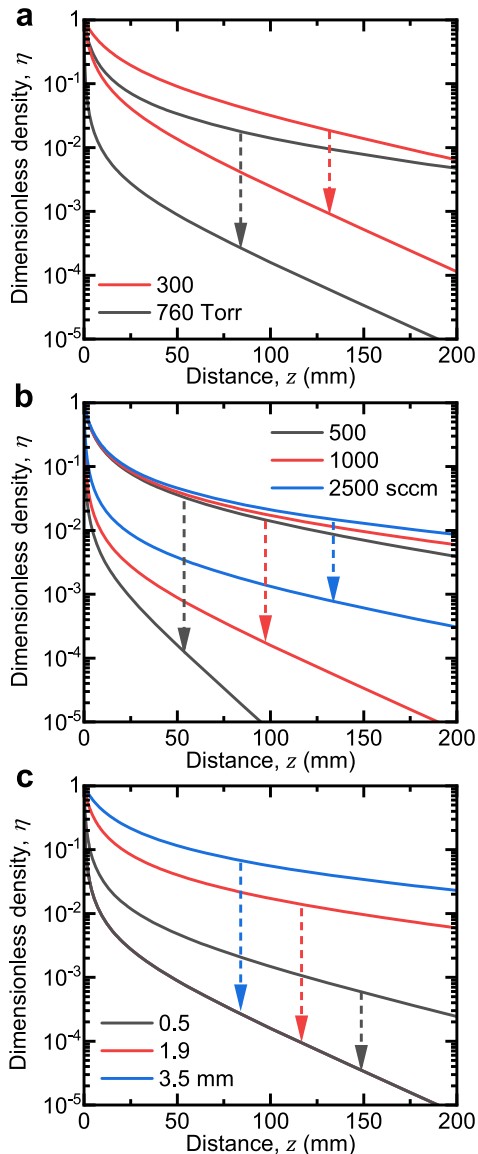

**Fig. 4 | Extension of spatial afterglow model to atmospheric pressure.**
**a** Dimensionless charge density, $\eta$, as a function of axial distance at the following constant parameters: $T_e = 1.5$ eV, $Q = 1000$ sccm, and $\Lambda = 1.91$ mm. In addition to atmospheric pressure, calculations at 75 and 300 Torr are shown for comparison.
**b** Dimensionless charge density, $\eta$, as a function of axial distance at different gas flow rates, $Q$, of 500, 1000, and 2500 sccm and the following constant parameters: $n_p = 4.1 \times 10^{20}$ m$^{-3}$, $T_e = 1.5$ eV, $Q = 1000$ sccm, and $\Lambda = 1.91$ mm. **c** Dimensionless charge density, $\eta$, as a function of axial distance at a tube inner diameter (characteristic diffusion length), $\Lambda$, of 1.18, 3.85, and 7.00 mm and the following constant parameters: $n_p = 4.1 \times 10^{20}$ m$^{-3}$, $T_e = 1.5$ eV, $Q = 1000$ sccm, and $\Lambda = 1.91$ mm. In all cases, background gas temperatures were assumed to be uniform and equal to one of two limiting conditions: an upper bound temperature at the plasma- afterglow boundary calculated using a heat transfer model, or a lower bound temperature of room temperature.

therefore, lessens the influence of gas temperature on charged species decay (Fig. 4c).

Gas temperatures in atmospheric pressure plasmas have been reported to vary from room temperature to ~1200 K[14,38,50–52]. Gas temperature affects both $\phi$ and $\lambda_c$, as indicated by the degree of curvature at small distances and the slope at large distances, respectively as shown in Fig. 4a (also see Supplementary Note 5). The Thiele modulus, $\phi$, decreases with increasing gas temperature due to lower neutral gas densities, and the characteristic decay length, $\lambda_c$, decreases due to

increasing volumetric flow rate. Within the range of typical values reported in the literature, our results show that volumetric flow rate (Fig. 4b) and tube inner diameter (Fig. 4c) can have significant effects on the decay profile. However, the effect of gas temperature is expected to be complicated by various factors, such as geometry, that can change the cooling rate significantly. Therefore, a thorough investigation of such factors is left for future work. However, in applications where an atmospheric pressure plasma jet is used as a source of charged or excited species to a surface, it appears that the chief parameters for determining the dose of said species are the distance from the plasma and the gas flow rate.

## Discussion

Plasmas are generally defined as a collection of oppositely-charged species, typically negatively-charged electrons and positively-charged ions, that have equal densities in the bulk, commonly referred to as quasi-neutrality. Electrons, because of their lighter mass, have higher electrical mobilities than ions. Two basic properties that can be derived from the properties of quasi-neutrality and electrical mobilities are the Debye length, $\lambda_D$ – which describes the distance over which any charge or potential is screened – and ambipolar diffusion coefficient, $D_a$ – which describes how electrons and ions diffuse collectively due to electrostatic forces. Only over distances corresponding to the Debye length, which is typically much smaller than the dimensions of the plasma, will the charge densities of electrons and ions not be equal.

In an afterglow, the charged species density decreases over time or space, which leads to fundamental changes to our picture of a plasma, as illustrated in Fig. 5a. Phelps previously showed that in a temporal afterglow, the apparent mode of diffusion changes from ambipolar diffusion to free diffusion[33]. The transition between the regimes was characterized by a normalized diffusivity, defined as the ratio of the apparent diffusivity to the intrinsic ion diffusivity, $D_{app}/D_i$, which in turn is a function of $\chi$, defined as the ratio of the characteristic diffusion length, $\Lambda$, and the Debye length, $\lambda_D$. Here, we applied a similar analysis to a spatial afterglow. We first used our model of the spatial afterglow to express $\lambda_D$ as a function of axial distance from the bulk plasma:

$$\lambda_D(\xi) = \lambda_{Dp}\sqrt{\exp(\xi) + \phi^2[\exp(\xi) - 1]} \tag{3}$$

where $\lambda_{Dp} = (\epsilon_0 T_e / q_e n_p)^{1/2}$ is the Debye length in the bulk plasma. Figure 5b shows that as the density of the charged species decays along the afterglow, $\lambda_D$ increases. At some point, $\lambda_D$ could even approach the dimensions of the reactor. At this extreme, we conjecture that the afterglow no longer follows the criteria for a plasma state.

By solving for the dimensionless distance, $\xi$, from Eq. 3, and following the methodology reported by Phelps[33], we calculated the normalized apparent diffusivity, $D_{app}/D_i$, for ions and electrons in the afterglow. Figure 5c shows that at small $\xi$, i.e., when $\chi$ is large (>1000), the normalized diffusivities for ions and electrons coincide and charged species follow ambipolar diffusion. At large $\xi$, where the charged species density decays to a much lower value, $\chi$ decreases, the normalized diffusivities for ions and electrons suddenly deviate (with the latter rapidly increasing and the former rapidly decreasing), and ions and electrons no longer diffuse at similar rates. For ions, the diffusion is completely free of electrostatic effects when $D_{app}/D_i$ is equal to unity, which is achieved at small $\chi$ values. Again, adopting Phelps' methodology[33], we can define a critical value for $\chi$ of 30, below which ambipolar diffusion no longer holds. This critical value occurs at different distances depending on the operating conditions, but always at the dimensionless distance $\xi_{crit}$, which can be expressed as:

$$\xi_{crit} = \ln\left[\frac{(\chi_p/\chi_{crit})^2 + \phi^2}{1 + \phi^2}\right] \tag{4}$$

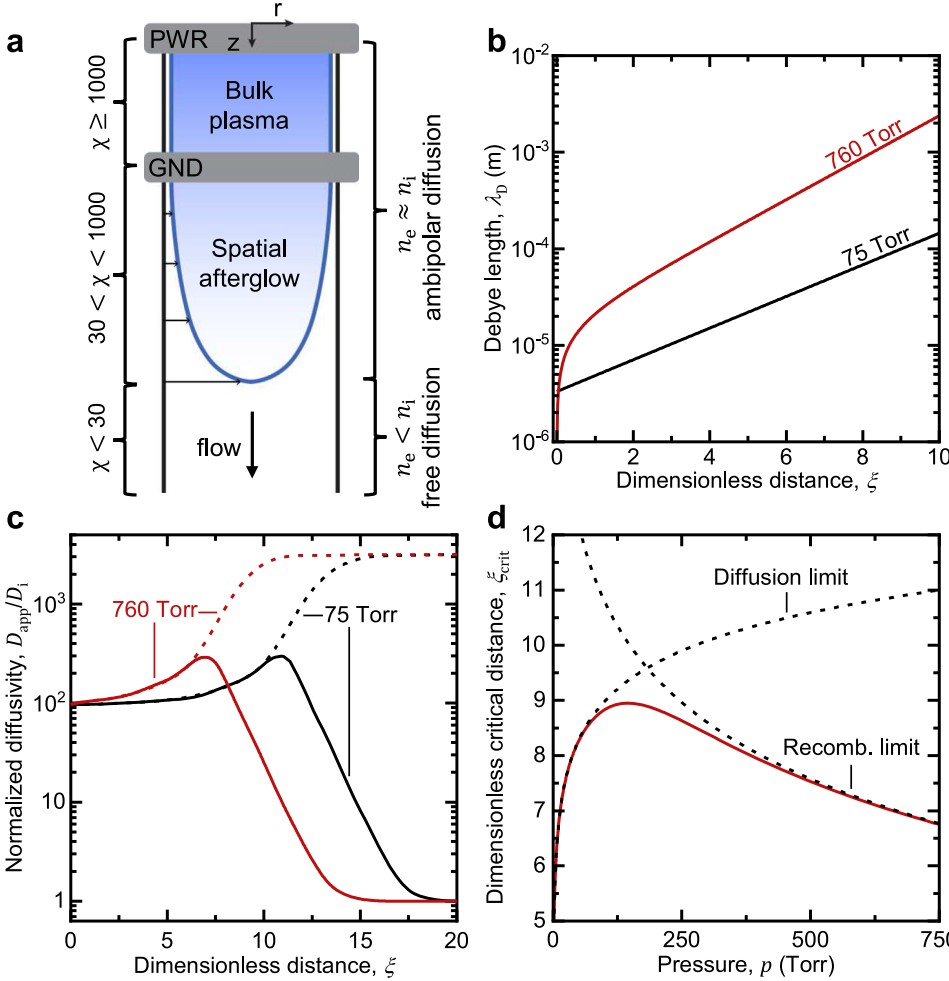

**Fig. 5 | Transition of diffusion regimes in spatial afterglow. a** Illustration of the growth of the Debye length (decrease in $\chi$) as charge decays in the spatial afterglow. **b** Debye length, $\lambda_D$, as a function of the dimensionless distance in the spatial afterglow, $\xi$, at 75 and 760 Torr. **c** Normalized diffusivity, $D_{app}/D_i$, as a function of dimensionless distance in the spatial afterglow, $\xi$, for ions (solid lines) and electrons (dashed lines) at 75 and 760 Torr. **d** Critical axial distance in the spatial afterglow, $\xi_{crit}$, for transition from ambipolar to free diffusion as a function of pressure (solid red line). $\xi_{crit}$ for the limiting cases of diffusion- and recombination-dominated charge density decay (dotted and dashed lines, respectively) are also shown for comparison (Supplementary Note 9). In all cases, the following parameters were kept constant: $T_e = 1.5$ eV, $T_g = 300$ K, $Q = 1000$ sccm, and $\Lambda = 1.91$ mm. In addition, $n_p$ was extrapolated from experimental measurements (Fig. 2a).

where $\chi_p$ corresponds to bulk plasma conditions and $\chi_{crit}$ is the critical value of 30. $\xi_{crit}$ increases and then decreases with pressure, exhibiting a peak at approximately 150 Torr (Fig. 5d). At pressures less than 150 Torr, the charge density in the bulk plasma is low, resulting in a relatively small value of $\chi_p$ (~2000), thus reaching the critical point at short distances from the plasma. At pressures above 150 Torr, the critical point is again reached at short distances from the plasma despite $\chi_p$ being relatively large (~10,000) due to the high rate of three-body recombination. The peak occurs because of opposing dependences on pressure: 1) bulk plasma density increases with pressure, and therefore, $\chi_p$ increases; 2) the characteristic time for recombination and therefore, $\phi^{-1}$, decreases with pressure (Supplementary Note 9). The interplay between bulk plasma density and the rate of charged species loss is important because sufficient decay (i.e., sufficient increase in the Debye length) must occur to reach the critical condition. Thus, higher plasma densities increase the critical distance, but at sufficiently high pressures, the faster rate of charged species decay decreases the distance corresponding to the critical condition. Remarkably, the peak at 150 Torr corresponds exactly to where modeling and experimental data of the afterglow showed some disagreement (Fig. 3a, c), and where experimental measurements began to show a significant drop in the charge density (Fig. 2b).

The consequence of the transition from ambipolar to free diffusion is that ions and electrons approach their intrinsic diffusivity limits, and because electrons are much more mobile, they are lost from the afterglow faster, leading to regions that are ion-rich and have a positive space charge. Thus, $\xi_{crit}$ demarcates a breakdown of ambipolar diffusion and quasi-neutrality. Such an environment could explain our DLP measurements of the afterglow at 400 Torr, which produced an atypical linear current–voltage trace (Supplementary Fig. 5). In support, the DLP measurement was performed approximately 80 mm downstream of the plasma-afterglow boundary, and we estimate $\xi_{crit}$ to be 8, which corresponds to 70 mm. We also carried out DLP measurements at this pressure at a distance below $\xi_{crit}$ and found that the measurements were more representative of DLP traces for a plasma (Supplementary Fig. 5). Thus, we suggest that the appearance of a linear trace is an indication of the absence of the plasma state and hot electrons.

Beyond the application of atmospheric-pressure plasma jets, the mathematical model developed in this work could potentially be extended to other environments characterized by charge decay. For example, in tokamak devices, divertor target plates are responsible for intercepting plasma species and impurities within the scrape-off layer (SOL) along the device walls[53,54]. High power fluxes can lead to sputtering, thermal stressing, and even melting of the plate material. For this reason, there is a need to reduce the flux or "detach" the plasma

from the plate[55]. Our model could be used to guide and understand how cold gas puffing or seeding leads to detachment by physical processes such as radiation, charge exchange, and recombination. We note that the model may need to be modified to account for a higher ionization fraction, which would introduce new terms such as ion/electron-mediated three-body recombination, and the externally applied magnetic fields that introduce restricted cross-field transport.

In summary, our study provides a comprehensive description of a spatial afterglow, including where the region begins, how charge decays within, and where it connects to the ambient. The dimensionless relation that was obtained for the charge density as a function of distance should be useful for many applications, such as predicting the order of magnitude of dose to a substrate in surface treatment or providing the charge density profile for the modeling of aerosol particle charging. The charge density can be relatively high and can contain hot electrons even when there is no light emission from the plasma, as observed through DLP measurements. At higher pressures or larger distances, our results show that the afterglow might not contain hot electrons if $z > \xi_{crit}\lambda_c$. Hence, our validated model could aid with predictions, guidelines, or experimental support for a wide range of applications of flowing plasmas, such as plasma jets and divertor detachment, in which the dynamics of charged species play a key role.

## Methods

### Experimental setup

The experiments were carried out with a continuous-flow plasma connected to a chamber where diagnostics were performed on the spatial afterglow (Supplementary Fig. 1). The plasma was formed inside a fused quartz tube (3.825 mm ID, 6.35 mm OD) between a powered steel ring electrode and a grounded, stainless-steel chamber. The quartz tube extended ~16 mm into the chamber, measured from the outside edge of the chamber. The ring electrode was electrically coupled to a RF power supply (13.56 MHz, Model RF-3, RFVII) in series with a homemade L-type matching network to minimize the reflected power, and an RF power meter (Octiv Poly, Impedans Ltd.). The diagnostic chamber was a six-way, stainless steel cube (CUBE-275, LDS Vacuum) with conflat flange ports. A rotary vane vacuum pump (RV-12, Edwards) was used to maintain vacuum within the system. The pressure was monitored using a capacitance manometer (Baratron 623H13TBE, MKS) and controlled using a needle valve. Argon (ultra-high purity, AR UHP300, Airgas) was introduced to the system using a mass flow controller (MCE-Series, Alicat Scientific). The standard flowrate and forward RF power for all experiments was 1000 sccm and 20 W, respectively. The plasma did not readily ignite above ~10 Torr at 1000 sccm argon flow rate, thus the plasma was first ignited at ~100 mTorr and an argon flow rate of 100 sccm. Once the plasma stabilized, the flow rate and the pressure were adjusted to desired conditions.

### Diagnostics

The bulk plasma density was estimated from electrical characterization by applying simple plasma-fluid and equivalent circuit models[38,39,56,57]. Briefly, the conduction current in the bulk plasma can be expressed by the drift-diffusion approximation as a function of the plasma density. The RF power meter was used to measure the electrical characteristics of the circuit. We approximate the plasma as a cylindrical conductor spanning between the two electrodes. Assuming that the drift contribution from ions is small compared to electrons due to their relatively lower electrical mobility and that the diffusive contributions to the current density are also negligible, the electron flux in the axial direction is related to the plasma density, $n_p$, as the following:

$$n_p = \frac{d}{Aq_e\mu_e R_p} \quad (5)$$

where $d$ is the electrode distance, $A$ is the cross-sectional area of the plasma, $q_e$ is the elementary charge, $\mu_e$ is the electron mobility, and $R_p$ is the resistance of the plasma. The resistance of the plasma was estimated by an equivalent circuit model where we approximated the plasma discharge as a capacitor and resistor in parallel (Supplementary Fig. 2)[58]. The plasma density obtained from the model represents a volumetric average and therefore, neglects any spatial or temporal variation within the bulk plasma. Details of the analysis are presented in Supplementary Note 1.

Double Langmuir probes (DLPs) were introduced in our experimental setup in the spatial afterglow chamber and positioned perpendicular to the direction of the flow. Two conflat flanges, one with a single hole in the center (referred to as the 'centerline') and one with two holes (one 10 mm and the other 6 mm from the centerline in opposite directions), were threaded to fit a 1/8" NPT to allow insertion of the DLP into the chamber, as illustrated in Supplementary Fig. 1. The two-hole flange could be rotated 180 degrees to obtain two more spatial positions for a total of five measurement positions along the axial direction of the afterglow. At least four DLP measurements were obtained for each of the spatial positions in the chamber at 10, 25, 50, 75, 150, 300, and 400 Torr. Supplementary Fig. 3 shows representative current-voltage curves measured at different pressures. Details of the analysis are presented in Supplementary Note 2.

## Data availability

The authors declare that the data supporting the findings of this study are available within the paper and its supplementary file, or from the corresponding author upon request.

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

## Acknowledgements
This work was supported by the Department of Energy under Grant No. DE-SC0018202 and the Air Force Office of Scientific Research under Grant No. FA9550-19-1-0088. We thank Prof. David Ruzic for insightful discussions.

## Author contributions

N.H.A. and R.M.S. conceived the study and designed the experimental setup. N.H.A. carried out the experiments with assistance from S.M.P. N.B.Ü. developed the Langmuir probe analysis. N.H.A. and N.B.Ü. analyzed the experimental results and constructed the analytical models. S.M.P. performed the Boltzmann equation calculations. NHA wrote the manuscript and prepared the figures. All authors helped revise the manuscript to its final form.

## Competing interests
The authors declare no competing interest.
