## [Peer Review File · Nature Communications]

Charge decay in the spatial afterglow of plasmas and its impact on diffusion regimesREVIEWER COMMENTS

Reviewer #1 (Remarks to the Author):

The authors perform a careful and in-depth analysis of the dynamics of a low-temperature plasma in its afterglow region, specifically for the case of a flow-through reactor. This particular geometry is of interest for the case of plasma jets impinging onto various substrates, either solid or liquid. Overall, this work is of the highest quality, especially its theoretical part. The authors do not use the “brute force” approach of a finite element plasma model, but rather perform a theoretical analysis that relies on simple but physically sound concepts (ratio of diffusivities, Debye length). The simplicity of the model will make it of great appeal to many researchers active in this area.

I do have a few comments that hopefully the authors can address in their revised manuscript:

- 1) The temperature dependence is quite interesting. I believe the authors assumed that temperature and pressure are independent. But most likely that is not the case. At higher pressure, both the increase in plasma density and in elastic collision frequency should lead to higher gas temperature. If the authors were to incorporate some pressure dependence on gas temperature, how would the results change?
- 2) Metastables play a crucial role in the afterglow. Late metastable-induced ionization helps extending the afterglow. They are not mentioned at all in the manuscript. At a minimum, the authors should comment on their expected effect on the afterglow.
- 3) The authors refer to “hot electrons” on multiple occasions. It would be good to define more

Reviewer #2 (Remarks to the Author):

The manuscript "Charge decay in the spatial afterglow of plasmas and its impact on diffusion regimes" by Abuyazid et al. depicted the use of a combined diffusion-recombination model to describe the spatial evolution of a plasma jet as it enters a diffusion chamber and it either diffuses towards the walls or becomes dissipated via recombination. The topic can be interesting considering recombination is also a critical issue in plasma detachment in tokamaks. However, the diagnostics is rather problematic and the issue, among with others, should be addressed before the paper can be further considered for publication.

The authors employed a double Langmuir probe to obtain the electron temperature and electron density using a model including ion-neutral collisional effects. However, at sufficiently high collisionality the electron retardation region of the I-V trace can also be affected by an energy dependent collision cross-section, vastly complicating the I-V trace. The authors should provide at least some evidences that this effect can be neglected.

In addition, DLPs fundamentally reflect only the information I-V trace of a single Langmuir probe near the floating potential, which can be distorted by either enhanced high energy tail or depleted high energy tail of the plasma EEDF. i.e. it works nicely only in a plasma with a strictly Maxwellian EEDF. Are the authors sure about having a single Maxwellian EEDF and why? It'll also be helpful if the authors provide an explanation why a single Langmuir probe was not employed instead.

The diffusion-recombination process, with which a plasma enters a highly collisional region, slowed down by collisions with neutrals and eventually recombines is common to plasma detachment in tokamaks. This is an important element that makes this article interesting (that it has value for two very distinct and wide groups of audiences) and the article is expected to be much better if the authors can give a good discussion on this commonality.

Response to Reviewers

Reviewer #1

The authors perform a careful and in-depth analysis of the dynamics of a low-temperature plasma in its afterglow region, specifically for the case of a flow-through reactor. This particular geometry is of interest for the case of plasma jets impinging onto various substrates, either solid or liquid. Overall, this work is of the highest quality, especially its theoretical part. The authors do not use the “brute force” approach of a finite element plasma model, but rather perform a theoretical analysis that relies on simple but physically sound concepts (ratio of diffusivities, Debye length). The simplicity of the model will make it of great appeal to many researchers active in this area.

I do have a few comments that hopefully the authors can address in their revised manuscript:

1) The temperature dependence is quite interesting. I believe the authors assumed that temperature and pressure are independent. But most likely that is not the case. At higher pressure, both the increase in plasma density and in elastic collision frequency should lead to higher gas temperature. If the authors were to incorporate some pressure dependence on gas temperature, how would the results change?

We thank the reviewer for raising the issue of the temperature and pressure interdependence. We believe, in particular, that the reviewer was referring to Figs. 4a and 4d, where results for the advection-diffusion-recombination model were shown separately as a function of pressure and temperature, which implied they are independent. As the reviewer pointed out, this may not necessarily be the case because as the pressure increases, the collision frequency increases and should lead to higher temperatures.

We fully agree with the reviewer’s point that the temperature and pressure should be correlated. However, we would like to clarify that there are two different regimes of relevance in our study. The first regime is the bulk plasma. Here, the plasma density is relatively high and there are elastic collisions (with higher frequency as the pressure increases). The second regime is the spatial

afterglow. Here, the plasma density decreases (more and more rapidly as the pressure increases) and the frequency of elastic collisions decreases. Thus, the temperature in the spatial afterglow, where we carry out the advection-diffusion-recombination model, and its dependence on pressure, will be a combination of heating (and cooling) mechanisms in the bulk plasma and mostly cooling mechanisms in the spatial afterglow.

To simplify this rather complex problem, we can separately analyze the bulk plasma and spatial afterglow regimes. Motivated by the reviewer's comment, we have now developed a heat model for the bulk plasma to estimate the temperature. Briefly, the model includes mechanisms for heat generation including elastic collisions and recombination, the latter of which is either in the volume (more important at higher pressures) or at the walls (more important at lower pressures). Through our analysis, we find that elastic collisions and wall recombination dominate heat generation, and volume recombination could be neglected. The model also includes mechanisms for heat loss by heat transfer to the walls, including radiation and natural convection. The model was solved at steady state in one dimension (1D) to obtain a radially-averaged temperature as a function of position, up to the end of the bulk plasma and the beginning of the spatial afterglow. Our results show, as expected by the reviewer, that the gas temperature at the beginning of the spatial afterglow increases with pressure.

We then carried out the advection-diffusion-recombination model at two different cases. First, the isothermal case assumes that the temperature in the spatial afterglow is the same as at the end of the bulk plasma and represents an upper bound. Second, the room temperature (300 K) case assumes that the temperature decreases rapidly and represents a lower bound. We note that the actual temperature in the spatial afterglow will be highly system-specific, depending on many factors, such as the geometry, materials used in the plasma source, and gas flow rate, which all affect the extent of cooling. The upper and lower bound illustrate the overall range of effects of temperature on the model results.

In revisiting our analysis in Fig. 4, we also realized that other parameters such as the gas flow rate and the tube diameter also affect the temperature, particularly in the bulk plasma. For this reason, we have now also applied our heat model to estimate the temperature in the bulk plasma at different

gas flow rates and tube diameters. We then again carried out the advection-diffusion-recombination model for the isothermal and room temperature cases.

Finally, motivated by the reviewer's comment, we also realized that our power model which we applied to the bulk plasma to estimate the boundary between the bulk plasma and the spatial afterglow could be refined by using the same 1D heat model we developed to estimate the temperature.

The following revisions have now been made to the manuscript and SI:

Manuscript, p. 13:

An important issue that must be considered as the pressure increases up to atmospheric is that the gas temperature in the plasma will increase as a result of the increasing electron-neutral collision frequency. We note that, because the charge density decreases in the spatial afterglow, the temperature in the plasma will not be the same as in the spatial afterglow, and it may decrease substantially depending on various conditions already discussed, such as pressure and distance, as well as other parameters, including gas flow rate and geometry. To address this issue, we developed a 1D heat transfer model for the bulk plasma to estimate the gas temperature at the end of the bulk plasma, i.e., the plasma-afterglow boundary. The description of the heat transfer model is detailed in the Supplementary Information. Briefly, the model includes mechanisms for heat generation, such as electron-neutral elastic collisions and recombination, in the volume as well as at the walls, including heat loss by radiation and natural convection to the walls. We then assume two limiting conditions for the temperature in the spatial afterglow. The first is isothermal, where the temperature in the spatial afterglow is the same as the plasma-afterglow boundary and represents an upper bound. The second is room temperature (300 K), where the temperature in the spatial afterglow is assumed to rapidly decrease to the ambient and represents a lower bound. The actual temperature in the spatial afterglow could be between these upper and lower bounds and will depend on the specific system; for this reason, we have purposefully left our analysis general.

Figure 4:

We have substantially revised Fig. 4. There was no reason to separately show temperature and pressure effects, and Fig. 4a and 4d have essentially been combined. Then, all of the analysis at different pressures, different gas flow rates, and different tube diameters, have been shown for an isothermal and room temperature case. We have also revised and added the following discussion to the manuscript pertaining to Fig. 4:

Manuscript, p. 13:

Figure 4a shows advection-diffusion-recombination model calculations for the decay of charged species at 760 Torr at an upper bound and lower bound temperature, where the former was calculated by the heat transfer model to be 1354 K. Results for 300 Torr are also shown for comparison where the upper bound temperature was calculated to be 655 K. We note that below 300 Torr, the upper bound temperature was found to be close to room temperature, consistent with results shown in Fig. 3c.

Figure 4. Dimensionless plasma density profiles, η , in the spatial afterglow as a function of axial distance at room and bulk plasma temperatures for a system at **(a)** 75, 300, and 760 Torr, keeping $T_e = 1.5$ eV, $Q = 1000$ sccm, and $\Lambda = 1.91$ mm. Similar plots at atmospheric pressure, spatial afterglows when **(b)** gas flowrate Q is 500, 1000, and 2500 sccm, **(c)** tube inner diameter

(characteristic diffusion length) Λ is 1.18, 3.85, and 7.00 mm. In (b) and (c), the default parameters are $n_p = 4.1 \times 10^{20} \text{ m}^{-3}$, $T_e = 1.5 \text{ eV}$, $Q = 1000 \text{ sccm}$, and $\Lambda = 1.91 \text{ mm}$. Arrows point from profiles at bulk plasma temperatures to room temperature, for each respective condition. For all isothermal cases depicted, plasma temperatures are given in Supplementary Table S3.

Manuscript, p. 14:

We also observe that, because the upper bound temperature increases with pressure, there is a larger possible span of charge density profiles (see Supplementary Table S3 and Fig. S9), but we note that gas temperature has the effect of reducing the decay of charge, serving to counter the effect of pressure.

Manuscript, p. 16:

Concomitantly, a higher gas flow rate leads to a lower bulk plasma temperature as a result of forced convection (see Fig. S10), which decreases the span of charge density profiles as compared to lower gas flow rates.

and:

Decreasing the tube diameter has minimal influence on the rate of charged species decay, but significantly enhances heat transfer (see Fig. 4c).

and:

Gas temperatures in atmospheric pressure plasmas have been reported to vary from room temperature to $\sim 1200 \text{ K}$.^{14,38,48–50} Gas temperature affects both ϕ and λ_c , as indicated by the degree of curvature at small distances and the slope at large distances, respectively (for example, see Fig. 4a). The Thiele modulus, ϕ , decreases with increasing gas temperature due to lower neutral gas densities, and the characteristic decay length, λ_c , decreases due to increasing volumetric flow rate. Within the range of typical values reported in the literature, our results show that volumetric flow rate (Fig. 4b) and tube inner diameter (Fig. 4c) can have significant effects on the decay profile. However, the effect of gas temperature is expected to be complicated by various factors, such as geometry, that can change the cooling rate significantly. Therefore, a thorough investigation of such factors is left for future work. However, in applications where an atmospheric pressure plasma jet is used as a source of charged or excited species to a surface, it appears that the chief parameters for determining the dose of said species are the distance from the plasma and the gas flow rate.

Manuscript, p. 8:

We invoked a power dissipation model to define the bulk plasma volume, which could be related to the boundary. Briefly, the power coupled to the bulk plasma is dissipated volumetrically by elastic collisions, three body recombination, and surface recombination on the walls. In our analysis, we found that the contribution of recombination was negligible compared to elastic collisions, so it was ignored. Solving the model yielded the volume and therefore, the length of the bulk plasma. The bulk plasma length could then be defined as the sum of length of the gap between the powered and ground electrode and the length of the plasma beyond the ground electrode, i.e., boundary, γ_0 . Qualitatively, with increasing pressure, γ_0 was found to decrease, indicating that the plasma becomes shorter; in other words, the plasma–afterglow boundary is located further upstream with pressure, as visually observed in Fig. 1.

Figure 3a:

Figure 3a. Distance from ground electrode of bulk plasma-spatial afterglow boundary, γ_0 , as a function of pressure, determined from experiments and calculated using a power dissipation model.

Heat transfer model of the bulk plasma. To extend our advection-diffusion-recombination model to high pressures (up to atmospheric), a 1D heat transfer model for the bulk plasma was constructed to estimate the gas temperature at the plasma-afterglow boundary.

We start by considering the spatial dependence of gas temperature in the plasma volume at steady state. We simplify our analysis by assuming that the gas temperature, T_g , is radially uniform and axially inhomogeneous. Potential mechanisms for heat generation include elastic collisions between electrons and neutrals and three-body recombination in the gas volume. Potential mechanisms for heat loss are convective heat transfer to the reactor walls. The 1D heat balance in the bulk plasma can thus be written as:

$$\dot{m}C_p \frac{dT_g}{dz} = \pi R_{iw}^2 \left(\Delta H_{rec} k_r n_g n_p^2 + \frac{2m_e}{M} \varepsilon_{mean} \nu_{el} n_p \right) - 2\pi R_{iw} h_{iw} (T_g - T_{iw}) \quad (\text{S31})$$

where \dot{m} is the mass flowrate, C_p is the specific heat capacity of the gas, ΔH_{rec} is the energy released from recombination, k_r is the three-body recombination rate coefficient, n_g is the neutral gas density, n_p is the plasma density, m_e is the mass of an electron, M is the mass of a neutral gas atom, ε_{mean} is the average kinetic energy of electrons, ν_{el} is the electron-neutral elastic collision frequency, h_{iw} is the convective heat transfer coefficient on the inner tube walls, R_{iw} is the tube inner radius (equivalent to R in the power dissipation and spatial afterglow model), and T_{iw} is the temperature of the inner wall. h_{iw} can be calculated from the Nusselt number, Nu , for a laminar flow in a cylindrical tube under constant temperature, assuming that the flow is thermally developed:¹⁸

$$Nu_{iw} = \frac{2R_{iw}h_{iw}}{k_{Ar}} = 3.66 \quad (\text{S32})$$

where k_{Ar} is the thermal conductivity of argon. This representation is valid if the inverse of the Graetz numbers are sufficiently large.¹⁸ To confirm this, we artificially varied the constant Nusselt number and found minimal effect on the results.

The inner wall temperature, T_{iw} , is unknown, but it can be determined by assuming steady state and equating the heat fluxes through the reactor walls. We considered the heat fluxes

on the inner walls (plasma facing side), within the quartz tube, and on the outer walls (in the ambient). On the inner walls, in addition to convective heat transfer from the gas, surface recombination of charges must be considered. Here, we adapt the rate of radial transport from the decay model and multiply it by the energy released from recombination to arrive at the heat generated. On the inner walls, the heat flux is:

$$\Gamma_{heat} = \frac{2\Delta H_{rec}D_a n_p}{R_i} + h_i(T_g - T_{iw}) \quad (S33)$$

where D_a is the ambipolar diffusion coefficient. Thermal conduction occurs through the solid reactor walls and depends on the difference between the inner and outer wall temperatures. However, we can evaluate the dimensionless Biot number to determine whether two separate wall temperatures are necessary. The Biot number compares the thermal resistance from convection in the gas phase to conduction in the solid phase. We found that the Biot number is significantly less than one up to 3000 K, allowing us to treat the problem with a singular wall temperature, T_{iw} .

On the outside of the cylindrical reactor, heat is dissipated through natural convection and thermal radiation. The heat flux on the outer wall is:

$$\Gamma_{heat} = \sigma_Q(T_{iw}^4 - T_\infty^4) + h_o(T_{iw} - T_\infty) \quad (S34)$$

where σ_Q is the Stefan-Boltzmann constant multiplied by the emissivity of quartz (0.93), h_o is the free convection heat transfer coefficient, and T_∞ is the ambient temperature (300 K). Similar to h_{iw} , h_o can be calculated from the Nusselt number, but the Nusselt number varies with the Rayleigh number, Ra , which is the ratio between thermal transport via diffusion to thermal transport via convection:

$$Nu_o = \frac{2R_o h_o}{k_{air}} = C(Ra)^n \quad (S35)$$

where C and n are empirical constants that depend on the value of Rayleigh number. The empirical left hand side of Equation S35 describes the Nusselt number for a vertical cylinder. The Rayleigh number is calculated by the expression:

$$Ra = \frac{16g\rho R_o^3 T_{iw} - T_\infty}{\mu\alpha T_{iw} + T_\infty} \quad (S36)$$

where g is the acceleration due to gravity, ρ is the ambient gas mass density, μ is the dynamic viscosity of air, and α is the thermal diffusivity of air.

Setting the heat flux on the inside of the tube (Equation S33) equal to the heat flux on the outside of the tube (Equation S34), the wall temperature, T_{iw} , can be determined as a function of the gas temperature, T_g , and the heat balance (Equation S31) can be solved as a simple initial value problem. Results are shown in Table S3 and Figures S9-S11.

Table S3. Gas temperatures calculated by the heat transfer model at the bulk plasma-spatial afterglow boundary for various pressures, gas flow rates, and tube (inner) diameters.

Pressure / Torr	Flowrate / sccm	Inner diameter / mm	Temperature / K
75	1000	3.85	342
300	1000	3.85	655
760	1000	3.85	1354
760	500	3.85	1670
760	1000	3.85	1354
760	2500	3.85	939
760	1000	1.18	310
760	1000	7.00	2070

Figure S9. (a-c) Calculated gas, inner wall, and outer wall temperatures; (d-f) volumetric heat transfer to the gas by elastic collisions and three-body recombination; (g-i) contribution of various heat flux mechanisms as a function of distance in the bulk plasma at 75, 300, and 760 Torr, respectively (1000 sccm Ar flowrate and 3.85 mm inner diameter). ‘Forced convection’ occurs in the tube; ‘Surface recombination’ occurs on the inner walls of the tube; ‘Solid conduction’ occurs across the tube walls; ‘Thermal radiation’ and ‘Free convection’ occurs from the outer walls to the ambient.

Figure S10. (a-c) Calculated gas, inner wall, and outer wall temperatures; (d-f) volumetric heat transfer to the gas by elastic collisions and three-body recombination; (g-i) contribution of various heat flux mechanisms as a function of distance in the bulk plasma at 500, 1000, and 2500 sccm, respectively (760 Torr and 3.85 mm inner diameter). ‘Forced convection’ occurs in the tube; ‘Surface recombination’ occurs on the inner walls of the tube; ‘Solid conduction’ occurs across the tube walls; ‘Thermal radiation’ and ‘Free convection’ occurs from the outer walls to the ambient.

Figure S11. (a-c) Calculated gas, inner wall, and outer wall temperatures; (d-f) volumetric heat transfer to the gas by elastic collisions and three-body recombination; (g-i) contribution of various heat flux mechanisms as a function of distance in the bulk plasma at 1.18, 3.85, and 7.00 mm inner diameter, respectively (760 Torr and 1000 sccm Ar flowrate). ‘Forced convection’ occurs in the tube; ‘Surface recombination’ occurs on the inner walls of the tube; ‘Solid conduction’ occurs across the tube walls; ‘Thermal radiation’ and ‘Free convection’ occurs from the outer walls to the ambient.

2) Metastables play a crucial role in the afterglow. Late metastable-induced ionization helps extending the afterglow. They are not mentioned at all in the manuscript. At a minimum, the authors should comment on their expected effect on the afterglow.

We would like to clarify that the focus of our study is a pure Ar plasma. We believe the reviewer's comment regarding metastables is in reference to gas mixtures (such as rare gas and rare gas-air mixtures), where there are metastables that are excited to relatively high energies and could then transfer that energy to another gas to ionize them, i.e., Penning ionization. For a pure Ar plasma, there should be no way for a metastable to impact ionization. That being said, we think it is an important point raised by the reviewer and could be relevant to future studies.

We have now made the following revisions:

Manuscript, p. 9:

We note that all experiments were performed in Ar and, to be consistent, the model also assumes only Ar as the background gas. Thus, ionization by mechanisms other than electron-impact such as Penning ionization were not considered.

Manuscript, p. 10:

We note that for plasmas formed in gas mixtures, the spatial afterglow would have additional Thiele moduli for each reaction that leads to charged species annihilation or generation, such as ion attachment and Penning ionization.

3) The authors refer to "hot electrons" on multiple occasions. It would be good to define more.

We thank the reviewer for pointing out the need to more clearly define the term "hot electrons". In general, hot electrons refer to electrons with energies substantially higher than the representative electron temperature, T_e , i.e. electrons that reside in the tail of the EEDF. Specifically for our study, the hot electrons are those that are collected by the double Langmuir probe and produce the electron retardation region of the I-V curve. Thus, the lack of a signature sigmoidal probe trace suggests the absence of hot electrons.

We have now defined our terminology in the manuscript on p. 13 as follows:

Specifically, at 400 Torr, the DLP traces did not exhibit the expected sigmoidal shape corresponding to the electron retardation region, and were instead linear, suggesting a low density of charged species and, in particular, the absence of hot electrons. Here, we define hot electrons as those with energies substantially higher than the background gas. Thus, the shift to a linear DLP trace could be interpreted as a transition to a free diffusion regime in which the hot electrons are rapidly lost from the afterglow.

Reviewer #2 (Remarks to the Author):

The manuscript "Charge decay in the spatial afterglow of plasmas and its impact on diffusion regimes" by Abuyazid et al. depicted the use of a combined diffusion-recombination model to describe the spatial evolution of a plasma jet as it enters a diffusion chamber and it either diffuses towards the walls or becomes dissipated via recombination. The topic can be interesting considering recombination is also a critical issue in plasma detachment in tokamaks. However, the diagnostics is rather problematic and the issue, along with others, should be addressed before the paper can be further considered for publication.

1) The authors employed a double Langmuir probe to obtain the electron temperature and electron density using a model including ion-neutral collisional effects. However, at sufficiently high collisionality the electron retardation region of the I-V trace can also be affected by an energy dependent collision cross-section, vastly complicating the I-V trace. The authors should provide at least some evidence that this effect can be neglected.

We thank the reviewer for this comment, and we agree that some details of the double Langmuir probe (DLP) measurements and analysis needed to be clarified. First, DLP has been previously applied to plasmas operating at high pressure where there is high collisionality, and similar drift-diffusion equations have been used to describe the ion and electron transport to the probe tips (Cozens & von Engel, *Int. J. Electron*, 1965). The theory has been validated in many experimental studies, including for both plasmas and flames (Wild *et al.*, *Contrib. Plasma Phys.*, 2012; Liu *et*

al., *J. Electrostat*, 2002). Second, we have now checked the validity for our own measurements. For all the pressures studied, the ratio of the electron mean free path to the Debye length is found to be at least two orders of magnitude greater than 1, which shows that the electrons can be assumed to be collisionless near the probe tips. Further, the electron retardation region of all the DLP traces we measured exhibited the characteristic sigmoidal shape (up to some distance within the spatial afterglow where there was a transition to free diffusion as discussed in our previous response to Reviewer #1).

We have now made the following revisions:

Supplementary Information, p. 4:

Theory shows that the characteristic sigmoidal shape of DLP traces observed in ideal, low-pressure systems is maintained at high pressures up to atmospheric, making it possible to apply drift-diffusion equations to describe ion and electron transport to the probe tips (Cozens & von Engel, *Int. J. Electron*, 1965). The theory has been previously validated by several experiments on plasmas and flames (Wild *et al.*, *Contrib. Plasma Phys*, 2012; Liu *et al.*, *J. Electrostat*, 2002), and in our case by calculating the ratio of the electron mean free path to the Debye length which was found to be $>10^2$ at all pressures studied, confirming that the electrons can be assumed to be collisionless.

Supplementary Information, p. 5:

The electron retardation region corresponds to the steep portion of the I - V trace and is analyzed to obtain the electron temperature. Collisionality between electrons and neutrals can complicate the analysis of this region by affecting the energy distribution of electrons. However, since the ratio of the electron mean free path to the Debye length is much greater than one (~ 3000 at 10 Torr and ~ 800 at 300 Torr), the electrons are effectively collisionless near the probe tips for the conditions considered in this investigation. Thus, the electron temperature can be extracted from the slope of this region where a small fraction of the electrons can reach the probe tips.

2) In addition, DLPs fundamentally reflect only the information I-V trace of a single Langmuir probe near the floating potential, which can be distorted by either enhanced high energy tail or depleted high energy tail of the plasma EEDF. i.e. it works nicely only in a plasma with a strictly Maxwellian EEDF. Are the authors sure about having a single Maxwellian EEDF and why? It'll also be helpful if the authors provide an explanation why a single Langmuir probe was not employed instead.

The reviewer is correct that in DLP, only electrons that are more energetic than the floating potential can be interrogated. However, the benefit of the floating circuitry is that the probe draws no net current from the plasma. We found that this is critical to performing reliable probe measurements in the spatial afterglow. The single Langmuir probe (SLP) always draws DC current from a plasma, and this is especially problematic for the spatial afterglow where the charge density is low and there is no electrical ground nearby. Moreover, at the relatively higher pressures we are studying, we found that the SLP can couple to the bulk plasma and draw large currents, which led to a filamentary discharge as shown below. In comparison, the DLP caused no perturbation at all pressures investigated and the measurements were highly reproducible.

Regarding the Maxwellian EEDF, as the reviewer suggested, in DLP analysis, we use the orbital-limited electron flux expression which assumes a Maxwell-Boltzmann distribution. We would like to further clarify that this assumption is self-consistent with our modeling approach. In our drift-diffusion model framework, we use an ambipolar diffusion description, $D_a = D_i(1+T_e/T_i)$ where D_a is obtained using the Einstein relation, which is derived on the grounds of Maxwell-Boltzmann statistics. For this reason, in this study, all of our analysis, from the model to the interpretation of I-V traces, has the assumption of a Maxwellian EEDF. We believe it is still a worthwhile first step to analyze the spatial afterglow assuming a Maxwellian EEDF.

To determine whether such an assumption is reasonable, we have now measured the electric fields in the bulk plasma and spatial afterglow. We then solved the Boltzmann equation for Ar using LoKI-B and the IST-Lisbon electron cross section database (Tejero *et. al. Plasma Sources Sci. Technol.*, 2019; Tejero *et. al. Plasma Sources Sci. Technol.*, 2021; Alves *J. Phys.: Conf. Series*, 2014) with the measured values of the electric field (i.e., reduced electric field, E/N) for the

bulk plasma and compared the EEDFs to Maxwellian and Druyvesteyn distributions. The calculations are shown in the figure below (Fig. S11) for 75 Torr and 300 Torr. We focus on the effective portion sampled by the DLP, which corresponds to electrons that can overcome the floating potential (~ 10 V) and substantially contribute to probe current. Our analysis shows that the EEDFs in this energy range are close to Druyvesteyn at lower pressures and higher E/N, and close to Maxwellian at higher pressures and lower E/N. However, these electrons will enter the spatial afterglow and gradually be lost due to diffusion and recombination, as observed by DLP measurements, which showed the disappearance of hot electrons at the ambipolar-to-free-diffusion transition. The electric field in the spatial afterglow is found to as expected monotonically decrease with distance to small values. Previous work has shown that the Boltzmann equation at low fields possesses a Maxwellian EEDF (Cherrington, *Gaseous Electronics and Gas Lasers*, 1979), and this is also shown in LoKI-B calculations at lower values of E/N. Additionally, recent studies of temporal afterglows of pulsed argon discharges revealed that deviations from a Maxwellian distribution was inversely proportional to the ionization fraction (Carbone *et. al. Plasma Sources Sci. Technol.*, 2015). Therefore, we conjecture that as the plasma decays and the field decreases, the EEDF will relax to a Maxwellian. Thus, the assumption of a Maxwellian EEDF at the locations we performed DLP measurements and applied our advection-diffusion-recombination model is acceptable and a useful simplification.

We have now made the following revisions:

Manuscript, p. 6:

Compared to single Langmuir probe (SLP), DLP has a floating circuit and, hence, withdraws no net current, which is critical to minimizing perturbation of the plasma and, in particular, the spatial afterglow, where the density of charged species is low, and there is no electrical ground nearby. Moreover, at higher pressures, SLP can couple to the plasma and draw large currents, which leads to a filamentary discharge whereby preventing the formation of the spatial afterglow as we ourselves observed (Supplementary Information, Fig. S6).

and:

During these measurements, the DLP caused no visible change to the spatial afterglow.

Figure S6. Photo of single Langmuir probe in spatial afterglow at 150 Torr showing coupling between the probe and the plasma which leads to severe change to the formation of the afterglow and damage to the tip as a result of heating.

Manuscript, p. 9:

In addition, we assumed that electrons exhibit a Maxwellian energy distribution in the spatial afterglow. Specifically, the Einstein relation was used to obtain the ambipolar diffusion coefficient, D_a , which is derived based on Maxwell-Boltzmann statistics. This assumption is self-consistent with our DLP analysis where the orbital-limited electron flux expression was used. In support, we measured the electric fields in the bulk plasma and spatial afterglow (Supplementary Information Tables S1 and S2), and calculated EEDFs at various conditions using LoKI-B (Tejero *et. al. Plasma Sources Sci. Technol.*, 2019; Tejero *et. al. Plasma Sources Sci. Technol.*, 2021). These calculations showed that the EEDF tended toward a Maxwellian at higher pressures and smaller values of the reduced

electric field, E/N (Supplementary Information Figure S8). Previous experiments have shown that the ionization degree is inversely proportional to a Maxwellian equilibrium (Carbone *et. al. Plasma Sources Sci. Technol.*, 2015), and calculations have shown that the solution to the Boltzmann equation for cooled electrons under low field conditions is a Maxwellian (Cherrington, *Gaseous Electronics and Gas Lasers*, 1979). Thus, the assumption of a Maxwellian is especially reasonable in the spatial afterglow, where electrons are lost and relax, and the electric field is low. Additional details of the electric field measurements and EEDF calculations are provided in the Supplementary Information.

Supplementary Information, p. 4:

DLP can directly measure the electric field strength in a plasma. As a result of the local electric field, current flows through the unbiased probes, and the symmetry center of the I - V curve is shifted away from the origin. Thus, the degree of shift in the x -axis of the curve gives the difference in plasma potential, and knowing the distance between the probes, the electric field strength is obtained.

Supplementary Information, p. 18:

Electric field in the bulk plasma and spatial afterglow. Similar to plasma density measurements, we performed measurements of the electric field in both the bulk plasma and the spatial afterglow, the latter being spatially resolved along the axial direction for a few conditions. The calculated values of the reduced electric field were then used as inputs for solving the two-term Boltzmann equation to obtain the electron energy density function (EEDF).

The electric field in the bulk plasma was estimated as a volume-averaged quantity by measuring the discharge voltage across the powered and ground electrode using the RF power probe. Briefly, the power probe measures the root-mean-square voltage, and we take this value to be the discharge voltage when the plasma is switched on, considering that stray components are in parallel to the plasma. Dividing the voltage by the distance between the powered and ground electrode, L_0 , yields the electric field, and further dividing by the gas number density, n_g , yields the reduced electric field, E/N .

We represent E/N in units of Townsends (Td) which is equal to $10^{-21} \text{ V m}^{-1}$. Table S1 a summary of E/N for the bulk plasma at several pressures.

The local electric field in the spatial afterglow was measured by rotating the DLP 90° such that the probe tips were aligned vertically (parallel to the flow) as opposed to the horizontal orientation (perpendicular to the flow) used to measure the plasma density. By doing so, the respective probe tips were in different vicinities in the afterglow along the axial direction—the direction of interest in this work—and sensitive to changes in the electric fields, which induce a voltage shift to the DLP traces (see Fig. S3 and Fig. S5). We determined this voltage shift by locating the point of symmetry in the DLP traces, and then estimated the electric field by dividing the voltage shift by the distance between the probe tips (1.5 mm). We stress that this method does not make any assumption about the energy distribution of electrons, nor the mechanism of ion flux to the probe (Cozens & von Engel, *Int. J. Electron*, 1965). Similar to the bulk plasma, the reduced electric field is calculated by dividing the measured electric field by the gas number density. Table S2 shows a summary of E/N as a function of axial distance in the spatial afterglow at 75 and 300 Torr.

Table S2. Reduced electric field calculated in the bulk plasma based on voltage measurements using the RF power probe. Gas temperature, T_g , was estimated using the heat transfer model (Equation S31), and distance between the electrodes was 2 cm.

Pressure / Torr	Voltage / V	Electric field / kV m^{-1}	Temperature / K	Reduced electric field / Td
10	390	19.5	300	60.6
75	415	20.8	342	9.79
150	460	23.0	432	6.86
300	415	18.0	655	4.06

Table S3. Reduced electric field in the spatial afterglow based on electric field measurements using DLP. Gas temperature, T_g , was assumed to be 300 K, and distance between the probe tips was 1.5 mm.

Pressure / Torr	Distance in the afterglow / mm	Electric field / $V\ m^{-1}$	Reduced electric field / $10^{-3}\ Td$
75	3.3	40.2	18.9
	3.9	30.7	14.4
	4.5	8.0	37.6
	4.9	8.0	37.6
300	6.5	35.4	8.0
	6.9	26.7	6.0
	7.5	20.5	4.6
	8.1	20.5	4.6
	8.5	1.1	0.24

Supplementary Information, p. 20:

Calculation of the electron energy distribution function based on the reduced electric field. Analysis of DLP measurements and the spatial afterglow decay model relied on the assumption that electrons in the afterglow exhibit a Maxwellian energy distribution. To support the existence of a Maxwellian electron energy distribution function (EEDF), we estimated the EEDF by solving a two-term electron Boltzmann equation at 75 Torr and 300 Torr. The measured reduced electric field (Table S1), plasma density (Figure 2a), gas temperature (Table S3), and excitation frequency (13.56 MHz) corresponding to the bulk plasma at these pressures, along with the electron–argon collision cross sections (including elastic, electronic excitation, and ionization) from the IST-Lisbon database (Alves *J. Phys.: Conf. Series.*, 2014), were used as input data for the code LoKI-B to calculate the EEDF within the bulk plasma (Tejero *et. al. Plasma Sources Sci. Technol.*, 2019; Tejero *et. al. Plasma Sources Sci. Technol.*, 2021). The computed EEDFs were compared to Maxwellian and Druyvesteyn distributions at identical values of E/N (Fig. S8). We focus only on electrons with energies at or below 20 eV, which are relevant to DLP measurements. In

general, we found that the EEDFs in this energy range appear to be closer to Druyvesteyn at lower pressures and higher E/N , and closer to Maxwellian at higher pressures and lower E/N . The EEDF in the spatial afterglow was not calculated because measurements of the plasma density relied on the assumption of a Maxwellian distribution. However, we note that as electrons enter the spatial afterglow, the increasing frequency of electron-neutral collisions (due to a decaying charge density) and the electric field decay (see Table S2) suggests that the EEDF will relax to a Maxwellian, which is consistent with previously reported solutions of the Boltzmann equation for cool electrons under low-field conditions (Cherrington, *Gaseous Electronics and Gas Lasers*, 1979). Additionally, recent studies of temporal afterglows of pulsed argon discharges revealed that deviations from a Maxwellian distribution was inversely proportional to the ionization fraction (Carbone *et. al. Plasma Sources Sci. Technol.*, 2015). Thus, our measurements and calculations show that the electrons in the spatial afterglow are indeed, or sufficiently close to, Maxwellian.

Figure S8. (a) Electron energy distribution function calculated for 75 Torr. (b) Electron energy distribution function calculated for 300 Torr. The reduced electric field, E/N , value used for each pressure corresponded to that calculated for the bulk plasma (see Table S2). Furthermore, the bulk plasma density (Fig. 2a) and gas temperature (Table S1) reported for the two conditions were also utilized. The excitation frequency was 13.56 MHz, corresponding to the RF power supply, and electron-electron collision effects were included.

3) The diffusion-recombination process, with which a plasma enters a highly collisional region, slowed down by collisions with neutrals and eventually recombines is common to plasma detachment in tokamaks. This is an important element that makes this article interesting (that it has value for two very distinct and wide group of audiences) and the article is expected to be much better if the authors can give a good discussion on this commonality.

We thank the reviewer for bringing to our attention the potential application of our model to the topic of plasma detachment at a divertor in tokamaks. Motivated by the reviewer's comment, we have now looked into strategies that have been reported to mitigate damage to the divertor plates, which fundamentally involve decreasing the plasma density/flux near the divertor. As suggested by the reviewer, we think that our mathematical model, which can treat the diffusion-recombination process, is applicable.

More specifically, the plasma approaching the divertor plates is partially analogous to the spatial afterglow. Near the divertor plate, an electron temperature of 5 eV is often used as a threshold for detachment, which is close to the properties of our spatial afterglow. In fact, recent models of plasma detachment have treated the problem as a plasma sheath. There are, however, a few modifications that would be necessary. In a fusion plasma, the ionization fraction and gas temperature will be much higher. Thus, additional recombination terms will need to be included in the model with ions and electrons as the third body instead of neutrals, and additional power dissipation and heat transfer terms may be necessary, most notably charge exchange and radiation.

We have now made the following revisions:

Manuscript, p. 20:

Beyond the application of atmospheric-pressure plasma jets, the mathematical model developed in this work could potentially be extended to other environments characterized by charge decay. For example, in tokamak devices, divertor target plates are responsible for intercepting plasma species and impurities within the scrape-off layer (SOL) along the device walls. (Krasheninnikov *et al.*, *Phys. Plasmas*, 2016; Leonard, *Plasma Phys. Control*

Fusion, 2018) High power fluxes can lead to sputtering, thermal stressing, and even melting of the plate material. For this reason, there is a need to reduce the flux or “detach” the plasma from the plate. (Stangeby, *Plasma Phys. Control Fusion*, 2018) Our model could be used to guide and understand how cold gas puffing or seeding leads to detachment by physical processes such as radiation, charge exchange, and recombination. We note that the model may need to be modified to account for a higher ionization fraction, which would introduce new terms such as ion/electron-mediated three-body recombination, and the externally applied magnetic fields that introduce restricted cross-field transport.

Sincerely,

REVIEWERS' COMMENTS

Reviewer #1 (Remarks to the Author):

I would like to thank the authors for their careful revision. I am fully satisfied with their answers!

Reviewer #2 (Remarks to the Author):

The authors had significantly improved the manuscript and has resolved all issues raised in my referee report. I'd recommend the publication of this manuscript.